# The Relationship between Physical Exercise and Negative Emotions in College Students in the Post-Epidemic Era: The Mediating Role of Emotion Regulation Self-Efficacy

**DOI:** 10.3390/ijerph191912166

**Published:** 2022-09-26

**Authors:** Shaohua Tang, Hanwen Chen, Lingzhi Wang, Tianci Lu, Jun Yan

**Affiliations:** College of Physical Education, Yangzhou University, Yangzhou 225127, China

**Keywords:** physical activity, negative emotions, anxiety, depression, emotion regulation self-efficacy

## Abstract

Objective: To investigate the relationship between physical activity and negative emotions among college students in the post-epidemic era and determine if emotional regulation plays a mediating role between physical activity and negative emotions. Methods: 479 college students (293 males, 186 females, M = 19.94, SD = 1.25) who were under closed campus management during the epidemic period were surveyed using the physical activity rating scale (PARS-3), the self-assessment scale for anxiety (SAS), the self-esteem scale for depression (SDS), and the emotion regulation self-efficacy scale (RES). Results: (1) Physical activity, negative emotions, and emotion regulation self-efficacy among college students were significantly different by gender (*p* < 0.01). (2) Physical exercise was negatively correlated with anxiety and depression (r = −0.236, *p* < 0.01; r = −0.198, *p* < 0.01) and positively correlated with emotion regulation self-efficacy (r = 0.256, *p* < 0.01) in college students. (3) Emotion regulation self-efficacy was negatively correlated with anxiety and depression (r = −0.440, *p* < 0.01; r = −0.163, *p* < 0.01). (4) Emotion regulation self-efficacy also partially mediated the relationship between physical activity and negative emotions. Conclusion: (1) Physical activity in the post-epidemic era negatively predicted anxiety and depression in school-isolated college students. (2) Emotion regulation self-efficacy in the post-epidemic era partially mediates the relationship between physical activity and anxiety and depression.

## 1. Introduction

In December 2019, novel coronavirus pneumonia (COVID-19) emerged in various parts of China and has since been highly proliferative and unpredictable, seriously endangering the physical and mental health of the public. The global outbreak is still at a high level, the virus is still mutating, and there is still great uncertainty about the final course of the epidemic [1]. For now, the overall situation of the epidemic in China is improving, but the risk of endogenous rebound of the domestic epidemic still exists, and the prevention and control of the epidemic on campus after students resume school needs to be given high priority [2]. Closed campus management is the status quo in epidemic prevention and control, and while students are kept safe and stable in this approach, being in a closed state for a long time may expose students to the risk of their needs not being met in time and cause fluctuations in their emotional states [3]. This causes a greater impact on the psychological health of college students and breeds many negative emotional problems, whereby depression and anxiety are among the more prominent ones, with serious cases affecting somatic health [4,5] and inadequate coping strategies [6]. General Secretary Xi Jinping has repeatedly highlighted the need to emphasize mental health and strengthen psychological guidance interventions [7]. Therefore, timely measures should be taken to address the anxiety and depression of college students under closed campus management during the new crown epidemic.

The psychological view of exercise is that physical activity can improve negative emotions, such as anxiety, depression, and frustration, giving individuals positive emotional experiences and improving mental health [8]. According to the theory of the temporal self-regulation of physical activity, many beneficial processes may be triggered by physical activity. For example, physical exercise may provide physiological benefits, while exercising with others or self-mastering an exercise contributes to mental health [9], and repeated physical activity often promotes other positive health behavior changes [10]. Relevant studies have shown that appropriate physical exercise can effectively alleviate state anxiety [11] and reduce depression [12,13]. During the normalization of the epidemic, the awareness of physical education among college students under closed management was strengthened, and the motivation and initiative to participate in physical exercise were increased to some extent [14]. Many studies have shown that moderate-intensity physical activity can effectively improve adverse states, such as depression and anxiety, and curb negative emotions among school students; the more active the level of physical activity commitment, the less negative emotions appear [15,16]. A longitudinal study confirmed that physical exercise was effective at improving anxiety and depression among home-isolated college students during the new coronary pneumonia epidemic [17]. However, the relationship between physical exercise, anxiety, and depression among isolated college students in schools under closed management during epidemic normalization is unclear. Based on this, this study proposes hypothesis 1: Physical activity in the post-epidemic era has a negative predictive effect on negative emotions among isolated college students in school.

Emotion regulation self-efficacy is an application of self-efficacy theory to emotion-regulation theory, which is developed based on emotion-related theories and refers to a degree of confidence that individuals have in managing their emotional states, which can reduce emotional tension, effectively alleviate anxiety and depression, and promote mental health [18,19]. According to Bandura’s (1977) self-efficacy theory, self-efficacy, as a core variable in an individual’s self-belief system, is an important resource and pathway for individuals to cope with trauma [20,21], and emotion regulation self-efficacy, as a self-efficacy with emotion regulation qualities, is (directly and indirectly) related to individuals’ negative emotions, such as anxiety and depression [22,23,24]. It has been suggested that emotion regulation self-efficacy may be an internal mechanism of physical activity that improves adverse states such as depression and anxiety [25]. It has also been confirmed that physical exercise helps to enhance the emotional regulation self-efficacy of college students, improves individuals’ anxiety and depression, and promotes mental health [26]. Emotion regulation self-efficacy plays an active role in both individuals’ ability to cope with unexpected emotional events and manage their own emotions [27]. Liu Yang et al. confirmed that emotion regulation self-efficacy could mediate the relationship between physical activity and negative emotions through a 5-week online exercise intervention with college students who were in home isolation during the epidemic [17]. However, such direct evidence is still lacking for university students under closed campus management during the Newcastle pneumonia outbreak. Based on this, this study proposed hypothesis 2: Emotion regulation self-efficacy mediates the relationship between physical activity and anxiety and depression.

In summary, this study systematically examines the relationship between physical exercise, emotion regulation self-efficacy, anxiety, and depression among college students under closed management during the new crown epidemic and the mediating role of emotion regulation self-efficacy between physical exercise, anxiety, and depression.

## 2. Materials and Methods

### 2.1. Participants

We conducted a cross-sectional study from April to May 2022 in Yangzhou City, Jiangsu Province, located in the eastern region of China. We used college students from a university in Yangzhou City, Jiangsu Province, China, as the research object and selected a university that implemented closed management during the normalization of the epidemic through a preliminary investigation and research. Data were obtained from 479 college students aged 17–26 (293 males, 186 females, M = 19.94, SD = 1.25). We used a QR code in our study that participants could scan with their cell phones to obtain the electronic questionnaire. A total of 593 questionnaires were recovered to ensure that the data collected were rigorous and valid; the recovered questionnaires were screened and sorted, excluding questionnaires that were completed too early, randomly answered, had the same answer for every question, or had incomplete answers. A total of 479 valid questionnaires were recovered, with a valid recovery rate of 80.8%. The study protocol was approved by the Ethics Committee of the Medical College of Yangzhou University.

### 2.2. Instruments

#### 2.2.1. Physical Activity Rating Scale

The Physical Activity Rating Scale (PARS-3), revised by Liang Deqing, was used [28]. The scale consists of 3 questions, which examine the amount of exercise in 3 dimensions: intensity, time, and frequency of participation in physical activity. Intensity and frequency were from 1 to 5 levels, respectively, 1 to 5 points; time was from 1 to 5 levels, respectively, 0 to 4 points, with the highest total score of 100 points and the lowest score of 0 points. Exercise assessment criteria: ≤19 as low motility; 20–42 as medium motility; ≥43 as large motility. Cronbach’s alpha coefficient for this scale in this study was 0.735.

#### 2.2.2. Anxiety Self-Assessment Scale

The “Anxiety Scale (SAS-CR)” developed by ZUNG was used [29], which is widely used to self-assess the level of anxiety among university students. The scale consists of 20 questions, 5 of which are reverse questions, using a 4-point scale, with the higher the total score, the higher the level of anxiety. Cronbach’s alpha coefficient for this scale in this study was 0.801.

#### 2.2.3. Depression Self-Rating Scale

The “Depression Self-Rating Scale (SDS)” developed by ZUNG was used [30]. The scale consists of 20 questions, 10 of which are reverse questions, using a 4-point scale, with a score below 50 being the normal range, with higher scores representing higher levels of depression in the individual. Cronbach’s alpha coefficient for this scale in this study was 0.719.

#### 2.2.4. Emotional Regulation Self-Efficacy Scale

The “Emotional Regulation Self-Efficacy Scale (RES)” developed by Caprara et al. was used, as modified by Wen Feng et al. [31]. The scale consists of 12 questions with a 5-point scale, and a higher total score indicates better ability. The scale is divided into 2 dimensions: self-efficacy for regulating positive emotions and self-efficacy for regulating negative emotions, which includes self-efficacy for regulating pain and frustration and self-efficacy for regulating anger and anger. Cronbach’s alpha coefficient for this scale in this study was 0.905.

### 2.3. Data Analyses

The collected data were subjected to descriptive statistics, independent samples t-test, Pearson correlation analysis using SPSS 26.0 for Windows(IBM, Armonk, NY, USA)and the bootstrap method to test for mediating effects.

## 3. Results

### 3.1. Control and Inspection of Common Method Deviation

Since we only used subjects’ self-reported answers to collect data in this study [32], the results may be affected by common method bias. Harman’s one-way test was used to examine the common method bias. Based on the unrotated factor analysis results, a total of nine factors with a characteristic root > 1 were extracted, and the explained variance of the largest factor was 30.11%, well below the critical criterion of 40%. Therefore, there was no significant common method bias in this study [33].

### 3.2. Participant Characteristics

A total of 479 college students participated in this study. The characteristics of the participants are shown in Table 1. Independent sample t-tests were conducted on three variables: physical activity, negative emotions, and self-efficacy for emotion regulation among school students of different genders. The results showed that there were significant differences in physical exercise, negative emotion, and emotion regulation self-efficacy, with male college students scoring higher than female college students in both variables of physical exercise and emotion regulation self-efficacy and lower than female college students in the variable of negative emotion.

### 3.3. Correlation Analysis among Variables

Pearson correlation analysis was performed for all variables (Table 2). The results showed that physical exercise was significantly and positively associated with emotion regulation self-efficacy (r = 0.256, *p* < 0.01). Physical exercise was also significantly and negatively associated with anxiety and depression (r = −0.236, *p* < 0.01; r = −0.198, *p* < 0.01). Emotion regulation self-efficacy was significantly and negatively correlated with anxiety and depression (r = −0.440, *p* < 0.01; r = −0.163, *p* < 0.01).

### 3.4. Analysis of the Mediating Effect of Emotion Regulation Self-Efficacy

In this study, the mediator model 4 in the SPSS plug-in-process prepared by Hayes was used [34]. Physical exercise was tested as the independent variable, negative emotion as the dependent variable, and emotion regulation self-efficacy as the mediating variable by substituting into the mediation model. The results showed that physical exercise significantly negatively predicted anxiety and depression (*p* < 0.01) and significantly positively predicted self-efficacy (*p* < 0.01); the negative predictive effect of physical activity on anxiety and depression remained significant (*p* < 0.01) after the inclusion of the variable of self-efficacy for emotion regulation. Thus, emotion regulation self-efficacy plays a partially mediating role between physical activity and anxiety and depression (Table 3 and Table 4).

The bootstrap method was used to test for mediating effects on a sample of 5000; the results showed that the confidence interval of the mediating effect of emotion regulation self-efficacy in the relationship between physical exercise and anxiety was [−0.160, −0.057], and the confidence interval of the mediating effect in the relationship between physical exercise and depression was [−0.072, −0.001], both excluding 0. That is, there was a mediating effect of emotion regulation self-efficacy in the relationship between physical exercise and negative emotions of college students. The values and percentages of direct and mediating effects are −0.132 (55.93%), −0.104 (44.07%); −0.167 (84.34%), and −0.031 (15.66%), respectively (Table 5 and Table 6; Figure 1 and Figure 2).

## 4. Discussion

### 4.1. Analysis of Physical Exercise and Negative Emotions of University Students under Closed Management during the Epidemic

The results of this study showed that male college students scored higher than female college students in physical activity when the campus experienced closed management during the epidemic, which is consistent with previous studies [26,35,36]. This is related to biological characteristics and the gender roles and personality traits assigned by society and culture; meanwhile, negative emotion scores are the opposite, which might be related to the fact that women tend to have greater emotional sensitivity but are more prone to mood swings and changes, while men tend to have lower emotional sensibility but more stable emotions [37,38]. As a result, women are more likely to develop negative emotions, such as anxiety and depression. By analyzing the indicators of the anxiety and depression self-assessment scale, we were able to conclude that the negative emotions of college students experiencing closed university management during the epidemic were concentrated in the following phenomena: more nervous and anxious than usual, easily angry; depressed mood, feeling tired for no reason; feeling scared and frightened for no reason, difficult to calm down; decreased sleep quality, insomnia, etc. This may be the result of a combination of factors: firstly, the epidemic on hand, whereby college students have been in a state of campus closure and control, and at the same time, they need to constantly receive various information about the epidemic, which may generate anxiety and negative emotions about not being able to enter and leave the campus freely during the epidemic, increasing the level of individual anxiety; secondly, students may feel depressed because if the campus is under constant closure and control, some of their needs cannot be met. In addition, college students are already facing academic, social, and employment pressures, which may be exacerbated by the campus’s closure during the epidemic.

### 4.2. The Relationship between Physical Activity and Negative Emotions and Emotion Regulation Self-Efficacy among University Students Isolated at School during the Epidemic

The results of the correlation analysis showed that physical activity significantly and negatively predicted anxiety and depression, and hypothesis 1 was valid. Participation in physical activity during the Newcastle pneumonia epidemic can improve individuals’ negative emotions, reduce stress, and improve college students’ mental health [39,40]. A longitudinal study found that physical activity significantly improved anxiety and depression among home-isolated college students during the epidemic [17]. It has also been confirmed that physical exercise can significantly and positively predict emotion regulation self-efficacy, indicating that physical exercise can be an effective means for individuals to enhance their self-efficacy during the epidemic. Moreover, it has positive and effective practical significance for maintaining the balance between positive and negative emotions and promoting the physical and mental health of college students, which is consistent with the findings of Wang Kun [38] and Wen Feng [31]. Self-efficacy was negatively correlated with negative emotions, indicating that self-efficacy can effectively mitigate and curb the expression of negative emotions, consistent with previous studies [41]. In addition, Bandura et al. showed that adolescents’ self-efficacy in regulating negative emotions was significantly associated with depressive tendencies [42]. In summary, emotion regulation self-efficacy may act as a protective factor that reduces anxiety and depression.

The results of the mediation effect analysis showed that emotion regulation self-efficacy partially mediated the relationship between physical activity and negative emotions, and research hypothesis 2 was tested. It was shown that individuals’ physical activity participation was positively related to emotion regulation self-efficacy, which in turn was indirectly negatively related to anxiety and depression. It has been shown that the direct effect of emotion regulation self-efficacy on anxiety and depression is greater than that of physical exercise and that the effect of physical exercise is mostly manifested through indirect effects [17], which is inconsistent with the results of this study. This may be due to the different geographical locations of the surveyed populations and the particular context of the new crown epidemic. The results of this study illustrate the mental health work of college students under closed campus management in the epidemic normalization stage, whereby students are encouraged to engage in physical exercise, actively carry out various sports activities, continuously improve their emotion regulation, and develop the ability to control negative emotions and alleviate or curb the expression of negative emotions.

There are also some limitations to this study. First, the measurements in this study were all online self-assessment data, which lacked objectivity, and future studies should construct more objective measurement indicators with the help of modern instruments and equipment. Second, this study is cross-sectional, which means it can be analyzed from a longitudinal perspective in future studies. Finally, this study analyzed only the total score for the variable of emotion regulation self-efficacy, which can be analyzed in a multidimensional way in future studies to further investigate the mediating role played by this variable.

## 5. Conclusions

Physical activity in the post-epidemic era negatively predicts anxiety and depression in school-isolated college students.

Emotion regulation self-efficacy in the post-epidemic era partially mediates the role between physical activity, anxiety, and depression.

## Figures and Tables

**Figure 1 ijerph-19-12166-f001:**
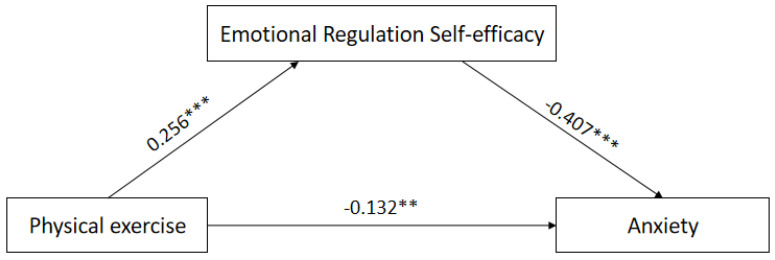
Mediated model effect map of emotion regulation self−efficacy on physical activity and anxiety. Note: ** means that *p*-value is <0.01, *** means that *p*-value is <0.001.

**Figure 2 ijerph-19-12166-f002:**
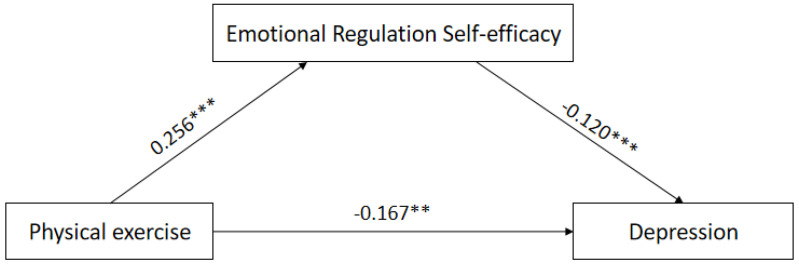
Mediated model effect map of emotion regulation self−efficacy on physical activity and depressed mood. Note: ** means that *p*-value is <0.01, *** means that *p*-value is <0.001.

**Table 1 ijerph-19-12166-t001:** Demographics and characteristics of the students participating in the survey.

Participant Characteristics	Total (*n* = 479)		Men (*n* = 293)		Women (*n* = 186)			
	M	SD	M	SD	M	SD	*t*	*p*
Age (year)	19.94	1.25	19.96	1.29	19.92	1.19		
Physical exercise	19.08	19.28	21.44	20.70	15.37	16.18	3.396 **	0.001
Anxiety	35.76	7.38	34.68	6.63	37.47	8.16	−4.097 ***	0.000
Depression	42.72	7.02	42.02	6.98	43.82	6.95	−2.759 **	0.006
Emotion regulation self-efficacy	43.04	10.26	44.15	10.23	41.30	10.09	2.977 **	0.003

Note: *n* = 479. ** means that *p*-value is < 0.01; *** means that *p*-value is < 0.001.

**Table 2 ijerph-19-12166-t002:** Results of the correlation analysis between physical activity, anxiety, depression, and self-efficacy for emotion regulation.

Variables	Physical Exercise	Anxiety	Depression	Self-Efficacy
Physical exercise	1			
Anxiety	−0.236 **	1		
Depression	−0.198 **		1	
Emotion regulation self-efficacy	0.256 **	−0.440 **	−0.163 **	1

Note: *n* = 479. ** *p* < 0.01.

**Table 3 ijerph-19-12166-t003:** A mediating model test of emotion regulation self-efficacy in physical activity and anxiety.

Variables	Emotion Regulation Self-Efficacy		Anxiety		Total Effect	
	*β*	*t*	*β*	*t*	*β*	*t*
Physical exercise	0.256	5.790 ***	−0.132	−3.140 **	−0.236	−5.315 ***
Self-efficacy			−0.407	−9.650 ***		
*R* ^2^	0.657		0.210		0.056	
*F*	33.522 ***		63.399 ***		28.247 ***	

Note: ** means that *p*-value is < 0.01, *** means that *p*-value is <0.001.

**Table 4 ijerph-19-12166-t004:** A mediating model test of emotion regulation self-efficacy in physical activity and depressed mood.

Variables	Emotion Regulation Self-Efficacy		Depression		Total Effect	
	*β*	*t*	*β*	*t*	*β*	*t*
Physical exercise	0.256	5.790 ***	−0.167	−3.613 **	−0.198	−4.402 ***
Self-efficacy			−0.120	−2.607 ***		
*R* ^2^	0.657		0.210		0.056	
*F*	33.522 ***		63.399 ***		28.247 ***	

Note: ** means that *p*-value is < 0.01, *** means that *p*-value is <0.001.

**Table 5 ijerph-19-12166-t005:** Mediating effects of emotion regulation self-efficacy in physical activity and anxiety.

Effect	Effect Value	Bootstrap 95% CI		Proportion of Relative Effect
		Boot LLCI	Boot ULCI	
Total effect	−0.236	−0.324	−0.149	
Direct effect	−0.132	−0.215	−0.050	55.93%
Indirect effect	−0.104	−0.160	−0.057	44.07%

Note: Boot LLCI and Boot ULCI refer to the standard errors and lower and upper 95% confidence intervals of the indirect effects estimated by the bias-corrected percentile bootstrap method, respectively. Indirect effect: physical activity → self-efficacy → anxiety.

**Table 6 ijerph-19-12166-t006:** Mediating effects of emotion regulation self-efficacy in physical activity and depressed mood.

Effect	Effect Value	Bootstrap 95% CI		Proportion of Relative Effect
		Boot LLCI	Boot ULCI	
Total effect	−0.198	−0.286	−0.110	
Direct effect	−0.167	−0.257	−0.076	84.34%
Indirect effect	−0.031	−0.072	−0.001	15.66%

Note: Boot LLCI and Boot ULCI refer to the standard errors and lower and upper 95% confidence intervals of the indirect effects estimated by the bias-corrected percentile bootstrap method, respectively. Indirect effect: physical activity → self-efficacy → depression.

## Data Availability

The data presented are available upon request from the corresponding author.

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
