# Peer review of "The Relationship between Physical Exercise and Negative Emotions in College Students in the Post-Epidemic Era: The Mediating Role of Emotion Regulation Self-Efficacy"

_ijerph, 2022, doi:10.3390/ijerph191912166_

Round 1
Reviewer 1 Report
This reviewer commends the authors for their study titled, “The relationship between physical exercise and negative emotions of College students in Post-epidemic Era: The Mediating role of emotion regulation self-efficacy.”
METHOD
1. The research hypothesis or research question(s) is/are not clearly articulated. The sampling type, sampling method, inclusion/exclusion criteria, and study population are not well defined.
2. Is Physical Activity an intervention instituted in this study? If so, what was the duration, intensity, and frequency of the intervention.
3. Are all participants subjected to the same duration, intensity, and frequency of the intervention.
4. This reviewer recommends full disclosure of the identity of the name and university affiliation of Institutional Review Board which provided ethical clearance for this study.
RESULTS
5. Study participants are not just numbers, they are people, human beings. Therefore, it is would be necessary to include demographic characteristics of the sample such as age, gender, years of education, occupation, income level, socioeconomic status, etc., and a descriptive/inferential analysis of how these demographic variable relate to the outcome variable.
6. For illustrations from Table 1 through 4, a narrative highlighting the statistically significant findings should accompany or replace the tables to help readers understand the study results.
Author Response
Dear editor and reviewer:
We have carefully studied the comments and observations given to us by the editors and reviewers and have done our best to revise them, and we apologies for the late response.
We have optimized the language in the introduction, discussion and conclusion sections and optimized the tables.
To reviewer 1:
Thank you for your comments and suggestions.
We have made changes in response to your comments.The following will respond to your comments article by article
Point 1:The research hypothesis or research question(s) is/are not clearly articulated. The sampling type, sampling method, inclusion/exclusion criteria, and study population are not well defined.
Response 1:We specify the research question or research hypothesis and add it clearly in the last part of the introduction(Line 83-86).We also clearly defined the sampling type, sampling method, inclusion/exclusion criteria, and study population(Line 92-102).
Point 2:Is Physical Activity an intervention instituted in this study? If so, what was the duration, intensity, and frequency of the intervention.
Point 3:Are all participants subjected to the same duration, intensity, and frequency of the intervention.
Response 2-3:Regarding your comments 2 and 3, this paper is a cross-sectional study, so there is no movement intervention.
Point 4:This reviewer recommends full disclosure of the identity of the name and university affiliation of Institutional Review Board which provided ethical clearance for this study.
Response 4:We have added the name and university affiliation of the institutional review board for ethical clearance and the number(Line 102,Line 288).
Point 5:Study participants are not just numbers, they are people, human beings. Therefore, it is would be necessary to include demographic characteristics of the sample such as age, gender, years of education, occupation, income level, socioeconomic status, etc., and a descriptive/inferential analysis of how these demographic variable relate to the outcome variable.
Response 5:We performed descriptive analyses of the demographic variables and recreated the tables for clarity of presentation(Line 139-147).
Point 6:For illustrations from Table 1 through 4, a narrative highlighting the statistically significant findings should accompany or replace the tables to help readers understand the study results.
Response 6:We recreated Table 1 with additional emphasis on statistically significant results. The following notes on Tables 1 to 4 clearly state their statistical significance(Line 147-148).

Reviewer 2 Report
Title: The relationship between physical exercise and negative emotions of College students in Post-epidemic Era: The Mediating role of emotion regulation self-efficacy
Article Type: Article
Summary
The present study examined the relationship between physical activity and negative emotions among college students in the post-epidemic era and also the mediating role of emotion regulation self-efficacy in relationship between physical activity and negative emotions. Participants were 479 college students who were included in this study and fulfilled some scales and questionnaires such as Physical Activity Rating Scale (PARS-3), the Self-Assessment Scale for Anxiety (SAS), the Self-Esteem Scale for Depression (SDS), and the Emotion Regulation Self-Efficacy Scale (RES). The results of this study indicated that physical activity in the post-epidemic era negatively predicted anxiety and depression in school-isolated college students and also emotion regulation self-efficacy in the post-epidemic era partially mediates the relationship between physical activity and anxiety and depression.
Suggestions
Pease add mean age ± SD of participants to the abstract.
P 3, L59. It is suggested to the authors first speak about Emotion regulation self-efficacy and then propose their hypothesis.
It is suggested to the authors speak about the basic theories regarding physical activity, emotion regulation self-efficacy and others in the introduction.
It is suggested that figure number 1 be included in the method section and not in the introduction.
It is suggested that the inclusion criteria to the study should be mentioned in the method section.
It is suggested that speak more about instruments and scales and also their methods for scoring. For example, about Physical Activity Rating Scale, how is grading? What is the range of the final score? How are different intensities of activity calculated?
Author Response
Dear editor and reviewer:
We have carefully studied the comments and observations given to us by the editors and reviewers and have done our best to revise them, and we apologies for the late response.
We have optimized the language in the introduction, discussion and conclusion sections and optimized the tables.
To reviewer2:
Point 1:Pease add mean age ± SD of participants to the abstract.
Response 2:Mean age ± SD has been added to the abstract(Line 16-17).
Point 2:.It is suggested to the authors first speak about Emotion regulation self-efficacy and then propose their hypothesis.
Response 2:Already added content on emotion regulation self-efficacy to the article(Line 65-72).
Point 3:It is suggested to the authors speak about the basic theories regarding physical activity, emotion regulation self-efficacy and others in the introduction.
Response 3:Theories related to physical activity and emotion regulation self-efficacy have been added to the introduction(Line 47-52,65-72).
Point 4:It is suggested that figure number 1 be included in the method section and not in the introduction.
Response 4:Figure 1 in the pre-revision article shows the hypothetical model diagram for this paper. In view of your suggestion, we have removed this hypothetical model diagram and attached clear assumptions(Line 83-86).
Point 5:It is suggested that the inclusion criteria to the study should be mentioned in the method section.
Response 5:We have added the inclusion criteria for the study in the methods section(Line 92-102).
Point 6:It is suggested that speak more about instruments and scales and also their methods for scoring. For example, about Physical Activity Rating Scale, how is grading? What is the range of the final score? How are different intensities of activity calculated?
Response 6:We have added more information about the scale, including the method of scoring(Line 105-110,112-115,117-120).

Round 2
Reviewer 1 Report
The authors appear to have addressed most of this reviewer’s concerns and comments.
Reviewer 2 Report
Thanks to the authors for revising the manuscript, according to the corrections were made, it seems that the present manuscript is acceptable for publication in the journal.